# Progress of the COVID-19: Persistence, Effectiveness, and Immune Escape of the Neutralizing Antibody in Convalescent Serum

**DOI:** 10.3390/pathogens11121531

**Published:** 2022-12-13

**Authors:** Dan Liang, Guanting Zhang, Mingxing Huang, Li Wang, Wenshan Hong, An’an Li, Yufeng Liang, Tao Wang, Jiahui Lu, Mengdang Ou, Zhongqiang Ren, Huiyi Lu, Rutian Zheng, Xionghui Cai, Xingfei Pan, Jinyu Xia, Changwen Ke

**Affiliations:** 1Guangdong Provincial Key Laboratory of Virology, Institute of Medical Microbiology, Jinan University, Guangzhou 510632, China; 2Emergency Key Team, Guangzhou National Laboratory, Guangzhou 510700, China; 3Guangdong Provincial Center for Disease Control and Prevention, Guangdong Workstation for Emerging Infectious Disease Control and Prevention, Guangzhou 511430, China; 4The Fifth Affiliated Hospital of Sun Yat-sen University, Zhuhai 519000, China; 5School of Public Health, Sun Yat-sen University, Guangzhou 510080, China; 6MPH Education Center, Shantou University Medical College, Shantou 515041, China; 7School of Public Health, Southern Medical University, Guangzhou 510515, China; 8Second People’s Hospital of Zhongshan, Zhongshan 528447, China; 9Huizhou Central People’s Hospital, Huizhou 516001, China; 10Department of Infectious Diseases, the Third Affiliated Hospital of Guangzhou Medical University, Guangzhou 510150, China

**Keywords:** kinetics, SARS-CoV-2, COVID-19, neutralizing antibody, SARS-CoV-2 variants, vaccination

## Abstract

Severe Acute Respiratory Syndrome Coronavirus 2 (SARS-CoV-2), a new coronavirus causing Coronavirus Disease 2019 (COVID-19), is a major topic of global human health concern. The Delta and Omicron variants have caused alarming responses worldwide due to their high transmission rates and a number of mutations. During a one-year follow-up (from June 2020 to June 2021), we included 114 patients with SARS-CoV-2 infection to study the long-term dynamics and the correlative factors of neutralizing antibodies (NAbs) in convalescent patients. The blood samples were collected at two detection time points (at 6 and 12 months after discharge). We evaluated the NAbs response of discharged patients by performing a micro-neutralization assay using a SARS-CoV-2 wild type. In addition, a total of 62 serum samples from discharged COVID-19 patients with Alpha, Beta, Delta, and Omicron variants of infection were enrolled to perform cross-neutralization tests using the original SARS-CoV-2 strain and VOCs variants (including Alpha, Beta, Gamma, Delta, and Omicron variants) and to assess the ability of NAbs against the SARS-CoV-2 variants. NAbs seroconversion occurred in 91.46% of patients (*n* = 82) in the first timepoint and in 89.29% of patients (*n* = 84) in the second detection point, and three kinds of NAbs kinetics curves were perceived. The NAbs levels in young patients had higher values than those in elder patients. The kinetics of disease duration was accompanied by an opposite trend in NAbs levels. Despite a declining NAbs response, NAbs activity was still detectable in a substantial proportion of recovered patients one year after discharge. Compared to the wild strain, the Omicron strain could lead to a 23.44-, 3.42-, 8.03-, and 2.57-fold reduction in neutralization capacity in “S_Alpha_”, “S_Beta_”, “S_Delta_”, and “S_Omicron_”, respectively, and the NAbs levels against the Omicron strain were significantly lower than those of the Beta and Delta variants. Remarkably, the NAbs activity of convalescent serum with Omicron strain infection was most obviously detectable against six SARS-CoV-2 strains in our study. The role of the vaccination history in NAbs levels further confirmed the previous study that reported vaccine-induced NAbs as the convincing protection mechanism against SARS-CoV-2. In conclusion, our findings highlighted the dynamics of the long-term immune responses after the disappearance of symptoms and revealed that NAbs levels varied among all types of convalescent patients with COVID-19 and that NAbs remained detectable for one year, which is reassuring in terms of protection against reinfection. Moreover, a moderate correlation between the duration of disease and Nabs titers was observed, whereas age was negatively correlated with Nabs titers. On the other hand, compared with other VOCs, the Omicron variant was able to escape the defenses of the immune system more significantly, and the convalescent serum infected with the Omicron variant played a critical part in protection against different SARS-CoV-2 variants. Recovery serum from individuals vaccinated with inactivated vaccine preceding infection with the Omicron strain had a high efficacy against the original strain and the VOCs variants, whereas the convalescent serum of persons vaccinated by inactivated vaccine prior to infection with the Delta variant was only potent against the wild-type strain.

## 1. Introduction

SARS-CoV-2, the pathogenic agent of COVID-19, is provoking a significant psychological, epidemiological, and economic burden all over the world [1]. The emerging variants, especially the highly contagious variants of concern (VOCs), accelerated the global spread of COVID-19 [2]. As of April 13, 2022, the disease has caused 499,119,316 confirmed cases and 6,185,242 deaths globally [3]. Immunity to SARS-CoV-2 natural infection was shown to provide a degree of protection against reinfection, and seropositive recovered patients were estimated to have 89% protection against reinfection [4,5]. The neutralizing antibodies (NAbs) elicited by infection are a central component of the immunity to subsequent challenges by SARS-CoV-2 and contribute a key effect for protective immunity against viral infections, and the virus neutralization test remains the laboratory gold standard for the detection of NAbs [6,7]. The duration of protective immunity was indefinite and the immune responses ineluctably attenuated according to different previous studies [5,8,9]. Multiple studies reported that NAbs titers in COVID-19 patients peaked within one month after symptom onset, subsequently reached a plateau, declined after the second month, persisted for 11 months, and still remained detectable for one year after infection [5,6,7,10,11,12,13].

Although SARS-CoV-2 variants were still globally diverse, VOCs progressively became the major epidemic strains. The Delta variant outbreak in Guangzhou in May 2021 was the first locally transmitted case in China, and the Delta variant subsequently became the superior variant with a tendency to substitute other variants [14]. The Omicron variant was first identified in Guangdong Province on 13 December 2021 and then gradually spread to become the predominant strain [15]. Similar to the Delta and Alpha variants, the Omicron variant was a highly divergent variant containing some concerning mutations associated with immune escape potential and higher infectivity, which led to inactivated vaccine showing less effectiveness against the Omicron strain [16,17]. However, vaccine inoculation still reduced the risk of hospital admission [18].

In this study, we characterized the kinetics of NAbs titers one year after SARS-CoV-2 infection by a micro-neutralization assay, and we also evaluated the span it took for recovery to become seropositive or return to seronegative. Simultaneously, we attempted to find out the factors affecting the NAbs titers and to explore the role of NAbs and inactivated vaccine inoculation in the convalescent serum from discharged COVID-19 patients recovering from Delta and Omicron variants against the key SARS-CoV-2 strains, including Alpha (B.1.1.7), Beta (B.1.315), Gamma (P.1), Delta (B.1.617.2), and Omicron (BA.1) variants.

## 2. Materials and Methods

### 2.1. Ethics Approval Statement

The protocol of the study was reviewed and approved by the Guangdong Provincial Center for Disease Control and Prevention (approval number: W96-027E-202121).

### 2.2. Serum Specimen

COVID-19 patients, after confirmed SARS-CoV-2 infection by a Real-Time Polymerase Chain Reaction detection (qRT-PCR), were sent to three sentinel hospitals in Guangdong province, China, which were appointed by the Guangdong provincial government to treat COVID-19, and were followed up for one year (from June 2020 to June 2021) after discharge with two virus neutralization assays. Demographic characteristics and clinical information were collected accordingly from the electronic medical record. We drew blood after patients recovered at two-time points, half a year and one year after discharge. Additionally, the serum samples from discharged COVID-19 patients infected with Alpha, Beta, Delta, and Omicron variants were subjected to cross-neutralization tests. The demographic characteristics and vaccination history were collected. The serum samples collected that were being processed in the Institute of Pathogenic Microbiology of Guangdong Provincial Center for Disease Control and Prevention (GDCDC), China, were de-identified prior to analysis and inactivated at 56 °C for 30 min. Furthermore, “duration of disease” refers to the period from onset to discharge. According to the Chinese clinical guidelines for the COVID-19 pneumonia diagnosis and treatment issued by the Chinese Health Council, all discharged patients met uniform discharge criteria, which were three consecutive days without fever, improvement in respiratory symptoms, significant resolution and recovery of acute lesions in lung image, and two negative results for nucleic acid test prior to discharge [19,20].

### 2.3. Vero-E6 Cells

Vero-E6 cells, an epithelial continuous cell line from the kidney of a normal monkey (*Cercopithecus aethiops*), were available from GDCDC (Guangdong, China). Vero-E6 cells were cultured in the growth media (GM), i.e., Minimum Essential Medium (MEM) (Gibco, Grand Island, NY, USA) which was supplemented with 1% 1M HEPES (Gibco, Grand Island, NY, USA), 1% 100 IU/mL Penicillin Streptomycin (PS) (Gibco, Grand Island, NY, USA), and 10% Fetal Bovine Serum (FBS) (Gibco, Grand Island, NY, USA). The cells were added into 96-well plates at the final concentration of 1.5 × 10^5^ cells per well, at 37 °C, in a 5% CO_2_ incubator for 24–48 h. The culture medium in plates with Vero-E6 cell monolayers was refreshed with the maintenance media (MM) prepared with MEM, 2% FBS, 1% HEPES, and 1% PS. The cells were for subsequent neutralization tests of SARS-CoV-2.

### 2.4. SARS-CoV-2 Isolates

The SARS-CoV-2 strains (wild type: No.20SF014 [21], Alpha strain: No.2021XG-131, Beta strain: No.2021XG-888, Gamma strain: No.2021XG-4123, Delta strain: No.2021XG-186, and Omicron strain: No.2021XG-5748) were isolated by the laboratory of the Institute of Pathogenic Microbiology of GDCDC. The viruses were titrated in serial dilutions (from 10^−1^ to 10^−8^) to get their median tissue culture infective dose (TCID_50_) on 96-well plates with Vero-E6 cell monolayers. The plates were inspected for the appearance of cytopathic effect (CPE) every day during the 7-day experiment by an inverted microscope, and the results were recorded. According to the Reed & Muench method [22], the final titers were calculated based on ten replicative wells of titration. All live-virus experiments were performed in a biological safety protection third-level laboratory using protocols approved by the Institutional Biosafety Committee. The comparison figure of the control and experimental group is displayed in Figure 1.

### 2.5. Virus Neutralization Test

Serum samples after inactivation, four-fold serial dilutions, starting from 1:4 with 60 μL per well (from 1:4 to 1:1024, 2 replicative wells per dilution), were then mixed into the plates with 180 μL/well MM and subsequently transferred to the neutralizing plates in the volume of 125 μL/well at each dilution for the following steps. The same volume of virus solution containing 100 TCID_50_ of SARS-CoV-2 viruses was then added to the 96-well plates. At the same time, the serum-control group and cell-control group were set up. The mixture was cultured for 2 h at 37 °C, in an incubator with 5% CO_2_. After incubation, a 100 μL/well mixture was added in sequence to the cell plates with sub-confluent cell monolayers of Vero-E6. The plates were cultured for 7 days at 37 °C in a 5% CO_2_ incubator.

After 7 days of culture, the plates were observed under an inverted microscope. The highest dilution that protected more than half of cells from CPE was regarded as the neutralization titer.

### 2.6. Statistical Analyses

Data were managed by using EpiData v3.1 (http://www.epidata.dk/, 20 July 2022). SPSS software v22.0 (https://www.ibm.com/cn-zh/analytics/spss-statistics-software, 20 July 2022) was used for descriptive analyses. All figures were drawn with GraphPad Prism software v8.0.2 (https://www.graphpad-prism.cn/, 20 July 2022) and R v3.6.1 (https://www.R-project.org/, 20 July 2022). Seroconversion was present as a change from seronegative (<1:4) to seropositive (≥1:4) [23]. All experiments were repeated three times independently.

Age variables were expressed using “Median and Interquartile Range ((IQR): 25–75%)”. The Chi-square test was performed to describe the comparison of categorical variables. Geometric Mean Titer (GMT) was used to represent the mean value of NAbs titers. Differences in NAbs titers by age, clinical classification, and duration of the disease were calculated through by analysis of variance or Fisher’s exact test, and the threshold levels of significance were adjusted for multiple comparisons by Bonferroni correction. The difference between the two infection groups in NAbs levels was examined by a student’s *t*-test. A two-tailed *p* value of <0.05 indicated statistical significance throughout the study.

## 3. Results

A total of 114 discharged patients were enrolled to find out the NAbs levels in a variety of patients with different symptoms, as described in the methods. Finally, 82 individuals completed the first test and 84 individuals completed the second test. Of these, 54 participated in the whole follow-up (Figure 2A1,A2).

### 3.1. Basic Information on Patients with COVID-19

The demographic and clinical characteristics of discharged patients with COVID-19 were shown in Table 1. This study included 56 (49.12%) males and 58 (50.88%) females, with a median age of 39 years (IQR, (27.75–50.25 years)). Eighteen (18/114, 15.79 percent) were asymptomatic cases and sixteen (16/114, 14.04 percent) were mild cases. A total of 80 out of 114 were moderate cases, without a statistically significant difference between the two detection time points (*p* = 0.170). The average duration of the disease was 23.09 days, and it in males was higher than in females (25.06 vs. 22.87). This may be attributed to sex steroids, sex chromosomes, and genomic and epigenetic disparities which may impair the immune response and thereby affect the response to SARS-CoV-2 infection [24,25,26,27,28,29,30].

Samples were collected at two detection time points for 114 patients, whereas some individuals were lost to follow-up at some time points and thus excluded from the following analysis (Figure 2B). In two tests, 82 of the 114 individuals completed the first test half a year after discharge, and NAbs remained detectable in the vast majority of patients (75/82, 91.46%) (Figure 2C). The 84 patients participated in the second test one year after discharge, of whom 10.71% (9/84) had undetectable levels of NAbs (Figure 2D). The GMT of the second interval was slightly lower than that of the first test (GMT_1st test_ vs. GMT_2nd test_, 1:23.21 vs. 1:17.09) (Table 1), and the seroconversion rate of NAbs decreased slightly with time. Moreover, positive percentages for NAbs were detected in more female patients than males after discharge in two detection intervals (Figure 2E). In addition, 54 survivors who participated in the whole follow-up period were included in the further analysis, with 26 (48.15%) males and 28 (51.85%) females, and with a median age of 36.50 years (IQR, (26.50–46.50 years)). Twelve (12/54, 22.22 percent) were asymptomatic cases, 10 were mild cases, and 32 of 54 were moderate cases. The average duration of the disease was 31.50 days. The proportion of patients with a seropositive NAbs titer (≥1:4) remained constant with time, and their NAbs became undetectable in 9.26% of patients at the second interval (Figure 2F).

### 3.2. Dynamics of NAbs Level to Discharged Patients

The 114 serum samples at two detection time points after discharge presented differences in the overall distribution of NAbs titers. As depicted in Figure 3A, NAbs titers were comparatively low at the second detection (12 months after discharge), but the value did not differ significantly between the first detection and the second detection (*p* = 0.098). The dynamics of SARS-CoV-2 NAbs progression in 54 patients who took both tests during the follow-up period were analyzed (Figure 3B). As illustrated in Figure 3C–E, the kinetics change of the patients in NAbs response was variable and can be divided into three categories. An unconverted dynamic curve evoked eight patients (14.81%), including six males and two females, who had no NAbs titers conversions, which developed on NAbs titer equal to 1:4 (No. 2) or equal to 1:8 (No. 1) or equal to 1:16 (No. 51–53) or equal to 1:32 (No. 15 and No. 46), or equal to 1:64 (No. 45). Three-quarters were asymptomatic cases (Figure 3C). The NAbs titers of the partial patients (19/54, 35.19%) increased one year after discharge and showed an uptrend curve in which NAbs titers further increased at the second detection time point (Figure 3E1–E5). This may be related to the vaccination of COVID-19 during this period. A previous study reported that the COVID-19 vaccine produced a partial immune response in the first dose, followed by a reassuringly protective immune response in the second dose, with an uptrend NAbs titer [31]. Among them, five individuals’ NAbs titers were negative half a year after discharge and reached 1:4 (No. 8 and No. 23) or 1:8 (No. 28) or 1:16 (No. 5) or 1:128 (No. 41) 12 months after discharge. The undetectable NAbs may be attributed to the need for persisting a second dose of vaccine booster, which is consistent with our study, and the NAbs are completely undetectable several months after infection due to individual differences in the levels [32,33,34].

However, more patients (27/54, 50%) emerged under a downtrend curve, in which NAbs titers reached a peak six months after discharge and then decreased thereafter (Figure 3D1–D7). Notably, the NAbs levels in five patients (9.26%) were found to decrease to negative at the time point of the second follow-up, including patients No. 3, No. 7, No. 9, No. 33, and No. 40. In addition, as shown in Table 2, there were statistical significances, comparisons of dynamic curves of different NAbs titers in age (*p* = 0.026), clinical classification (*p* = 0.003), and duration of disease (*p* = 0.015).

### 3.3. The Influencing Factors of NAbs Level in COVID-19 Convalescent Patients

To study the changes and influencing factors of NAbs response according to age, gender, disease severity, and duration of the disease, we performed analyses of stratified subgroups in 54 convalescent patients who participated in the whole follow-up period. NAbs titers for SARS-CoV-2 did display a statistically significant change between the age of <18 and 18–50 in the second detection (*p* = 0.011; Appendix A). NAbs titers in the group of people with mild or moderate symptoms at 6 months discharge were non-significantly high compared with the group of patients at 12 months discharge (*p* = 0.205 and *p* = 0.170, respectively; Figure 4C). Furthermore, no differences were observed for age, gender, duration, or disease severity at the two detection time points (Figure 4A,B,D).

We also assessed the dynamics of the patient’s age and disease duration development and attempted to ascertain whether the appearance of NAbs responses was connected with the age and duration of the disease (Figure 5). A correlation analysis showed a moderate correlation between the duration of disease and NAbs titer (R = 0.324, *p* = 0.0168, Figure 5B), whereas the age (R= −0.104, *p* = 0.450, Figure 5A) showed a slightly negative correlation with NAbs titer.

### 3.4. The NAbs against Other Variants in COVID-19 Convalescent Patients

To further study the roles of the NAbs and vaccine inoculation against other SARS-CoV-2 variants, a total of 62 convalescent serum samples from discharged COVID-19 patients were enrolled, including serum samples after Alpha, Beta, Delta, and Omicron variants infection, named “S_Alpha_”, “S_Beta_”, “S_Delta_”, and “S_Omicron_”, respectively. We performed cross-neutralization tests using the wild type, Alpha, Beta, Gamma, and Delta variants, that is “V_Wild_”, “V_Alpha_”, “V_Beta_”, “V_Gamma_”, “V_Delta_”, and “V_Omicron_”, accordingly. The vaccination history was collected, and the cross-reactive NAbs titers were calculated. The demographic characteristics, vaccination histories, and GMTs of discharged patients with COVID-19 were shown in Table 3. The median age was 38 years in the S_Alpha_ group, 47 years in the S_Beta_ group, 44 years in the S_Delta_ group, and 35 years in the S_Omicron_ group, respectively. Two (66.67%) participants in the S_Alpha_ group, three (100%) participants in the S_Beta_ group, twelve (38.71%) participants in the S_Delta_ group, and four (16%) participants in the S_Omicron_ group were male. More than one-third of subjects reported a history of non-vaccination, and 14 out of the 62 participants were immunized with three doses of inactivated vaccine inoculation pre-infection.

Compared to the V_Wild_, the V_Omicron_ could lead to a 23.44-, 3.42-, 8.03-, and 2.57-fold reduction in neutralization capacity in “S_Alpha_”, “S_Beta_”, “S_Delta_”, and “S_Omicron_”, respectively. Notably, the NAbs against V_Omicron_ exhibited a significant fold reduction compared to V_Beta_ and V_Delta_. It was the most immune escape variant among the six strains analyzed. Nevertheless, the GMTs of S_Omicron_ against V_Wild_, V_Alpha_, V_Beta_, V_Gamma_, V_Delta_, and V_Omicron_ were 263.55, 256.34, 164.50, 572.82, 223.16, and 102.67, respectively. Perhaps the NAbs from convalescent serum infected with the Omicron variant played an essential role against different SARS-CoV-2 variants, which is, in principle, comforting in terms of protection against reinfection.

As shown in Table 4, the results preliminarily explored the impact of the vaccination prior to infection by NAbs titer determination. Although vaccination induced both humoral and cellular responses, vaccine-induced NAbs were a persuasive protection mechanism against SARS-CoV-2 [35]. For individuals recovering from Delta variant infection, only the GMT after one dose of inactivated vaccine inoculation was higher than those of non-vaccinated individuals when using the V_Wild_ to perform the cross-neutralization test (*p <* 0.05). In participants recuperating from Omicron variant infection, the GMTs of persons who received two or more doses of inactivated vaccine were found to be significantly higher than in those who were non-vaccinated and received the first dose of vaccine when using V_Wild_, V_Alpha_, V_Gamma_, V_Delta_, and V_Omicron_ to perform cross neutralization tests (*p* < 0.05). The result of NAbs titers in the cross neutralization test was presented in Appendix A. Regardless of the SARS-CoV-2 strains analyzed for the cross-neutralization test, the NAbs titers were significantly higher in S_Omicron_ than S_Delta_ (*p* < 0.05), but notably lower in S_Omicron_ than in V_Gamma_ for V_Delta_ and V_Omicron_. The NAbs level in the S_Delta_ against V_Delta_ was significantly more effective than in V_Wild_, V_Alpha_, V_Beta_, and V_Omicron_ (*p* < 0.05).

## 4. Discussion

Our results demonstrated that convalescent plasmas post-infection with SARS-CoV-2 were immunogenic, with seroconversion rated over 90% at 6 months post-discharge and nearly 90% at 12 months post-discharge. We also estimated the kinetics of NAbs titers for 54 recovery patients at 6 months and 12 months post-discharge. The NAbs dynamics curves of the patients were divided into three categories, and the relevant influencing factors were preliminarily investigated.

Given the limitations of our research method, our results only conservatively demonstrated that the NAbs remained detectable in most survivors, although a slight decrease in NAbs titers was found in a fraction of individuals one year after discharge. It is similar to the previous description in SARS-CoV studies [11,36,37,38]. This finding is comforting because NAbs play a critical role in protection against SARS-CoV-2 infection. Additional studies are needed to support it and to further understand the significance and dynamics of the cellular immunity and humoral immunity function to SARS-CoV-2, including the duration of memory B cells and plasma cells in vivo [39,40,41] and the continuance of SARS-CoV-2-specific IgG and IgM antibody levels in patients recovering from COVID-19 [42]. Particularly striking was the fact that the majority of the 54 convalescent patients had NAbs titers falling below 1:32 12 months after discharge, raising some concerns about whether such low NAbs responses would be sufficient to fully protect individuals from reinfection, although there were currently no widely accepted cut-off values, and a negative or weakly positive result for NAbs titers did not necessarily imply a lack of protection ability [32]. Conspicuously, the NAbs titers in five out of 54 patients (9.26%) dwindled after the originally seropositive results and obtained seronegative titers at the second detection time point during the follow-up, which was also described in other studies delineating individual differences in NAbs persistence after natural infection of SARS-CoV-2 [43,44,45]. It was worth noting that eight survivors showed an unconverted dynamic curve, i.e., a development that neither rose nor fell. This may be associated with individuals’ pre-infection health status, such as immune efficacy [46].

We preliminarily analyzed NAbs responses stratified according to age, gender, disease severity, and duration of the disease. It is noteworthy that age was a factor affecting NAbs levels, which slightly discounted with age, which was consistent with previous studies [33,47], whereas NAbs titers moderately raised with the duration of the disease. Although no statistically significant differences in NAbs titers were observed between patients of different genders, females had slightly higher NAbs titers values than males. The reason for this result may be bound up with differences in hormone levels, as with Estradiol (E2), which boosted the number of antibody-secreting cells [48,49,50,51].

In addition, to better treat COVID-19 patients in clinical practice, the National Health Commission of the People’s Republic of China incorporated recovery plasma therapy into the Trial 8th Edition diagnosis and treatment program of Novel coronavirus pneumonia. Based on this, NAbs titers in two tests were artificially divided and analyzed according to their trend in order to provide a foundation for the screening of donors and the collecting of convalescent plasma with higher NAbs titers [32,52].

We also investigated whether serum samples from COVID-19 patients who had recovered from Alpha, Beta, Delta, and Omicron variants infection would be able to neutralize the original strain and VOCs variants, and analyzed whether these serum samples from individuals who had been vaccinated pre-infection would provide a protection mechanism against SARS-CoV-2. Using a cross-neutralization assay, we found that NAbs titers in serum samples infected with Alpha, Beta, Delta, and Omicron against the Omicron strain were low, and the serum neutralized the Omicron strain to a much lesser extent than any other variants analyzed (such as the Wild-type, Alpha, Beta, Gamma, and Delta strains), which was the same as the previous study of the Omicron variant [53]. The serum samples sourced from Omicron convalescent-vaccinated persons were able to neutralize the SARS-CoV-2 variants analyzed to a greater degree than the Delta strain, which was inconsistent with the previous study [53]. Our study showed a significant increase in vaccine-induced NAbs levels after Delta and Omicron infection in vaccinated participants compared with unvaccinated persons, as in the earlier study [54].

There are a few limitations to this study. First, although we intended to include discharge patients diagnosed in three sentinel hospitals in Guangdong province, China during the research period, some patients were inevitably lost to follow-up. Furthermore, except for basic demographic characteristics (such as age and gender), clinical classification, and hospitalization duration, we lacked information on all patients’ other demographic information, such as occupation, body mass index, etc., and clinical data, including clinical symptoms and signs, underlying diseases, and treatment needs (oxygen supplement requirement), and even biochemical parameters and hematologic markers, such as the hormone mediators that also affected NAbs responses (like serum amyloid A (SAA) or E2 levels) and lymphocyte count level, etc. [33]. All of the above resulted in our inability to determine the correlation between other factors and the convalescent patients’ NAbs response. Some findings were previously found in other research that although SARS-CoV-2 variants were still globally diverse, VOCs progressively became the major epidemic strains, with a tendency to substitute other variants, such as the superior variant in 2021—the Delta variant [55,56,57]. With the exception of the Beta and Alpha variants, the wild-type virus was the prevalent strain between June 2020 and June 2021. It was not until 6 January 2021 and 20 May 2021 that the new variants (Beta and Delta) emerged in Guangdong province, respectively [14,58]. Taking the above reasons into account, we restricted the use of the original SARS-CoV-2 (wild type) and Beta variant for micro-neutralization assays because of premature follow-up times (from June 2020 to June 2021). Due to the instability of the titration results in the Beta variant during the experiment, we subjectively discarded these data and did not analyze them. Obviously, our immediate problem was whether convalescent patients possessed the capacity to neutralize emerging VOCs. Therefore, we conducted another study to try to remedy this defect. We performed the live virus cross-neutralization assay to obtain information on NAbs titers against the new VOCs, such as the Delta and Omicron variants.

## 5. Conclusions

In conclusion, our findings threw highlighted the dynamics of the long-term immune responses after the disappearance of symptoms, that NAbs levels varied among all types of convalescent patients with COVID-19, and that NAbs remained detectable for one year, which is reassuring in terms of protection against reinfection. Moreover, a moderate correlation between the duration of disease and Nabs titers was observed, whereas age was negatively correlated with Nabs titers. On the other hand, compared with other VOCs, the Omicron variant was able to escape the defenses of the immune system more significantly, and the convalescent serum infected with the Omicron variant played a critical part in the protection against different SARS-CoV-2 variants. Recovery serum from individuals vaccinated with inactivated vaccine preceding infection with the Omicron strain had a high efficacy against the original strain and the VOCs variants, whereas the convalescent serum of persons vaccinated by inactivated vaccine prior to infection with the Delta variant was only potent against the wild-type strain.

## Figures and Tables

**Figure 1 pathogens-11-01531-f001:**
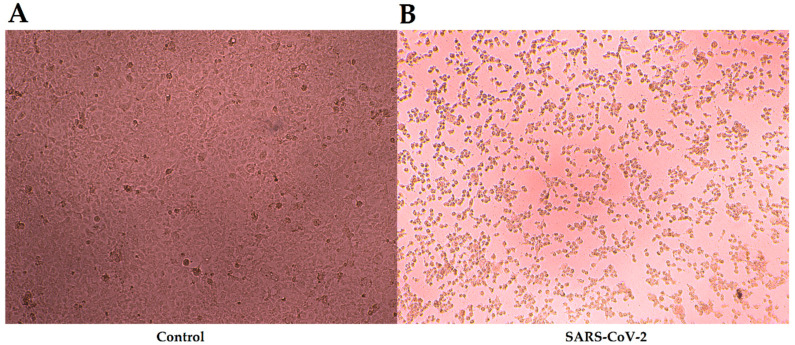
CPE induced by 20SF014 isolates (wild type) of SARS-CoV-2 strain in Vero-E6 cells. CPE refers to the occurrence of changes in cells inoculated with the SARS-CoV-2 strain, such as cell swelling, widening of cell gaps, loss of intercellular junctions, the appearance of multinucleated giant cells, high refractive index, and floating of partial cells [21]. (**A**) Normal Vero-E6 cells. (**B**) Vero-E6 cells infected with wild type of SARS-CoV-2 strain. CPE: cytopathic effect; SARS-CoV-2: Severe Acute Respiratory Syndrome Coronavirus 2.

**Figure 2 pathogens-11-01531-f002:**
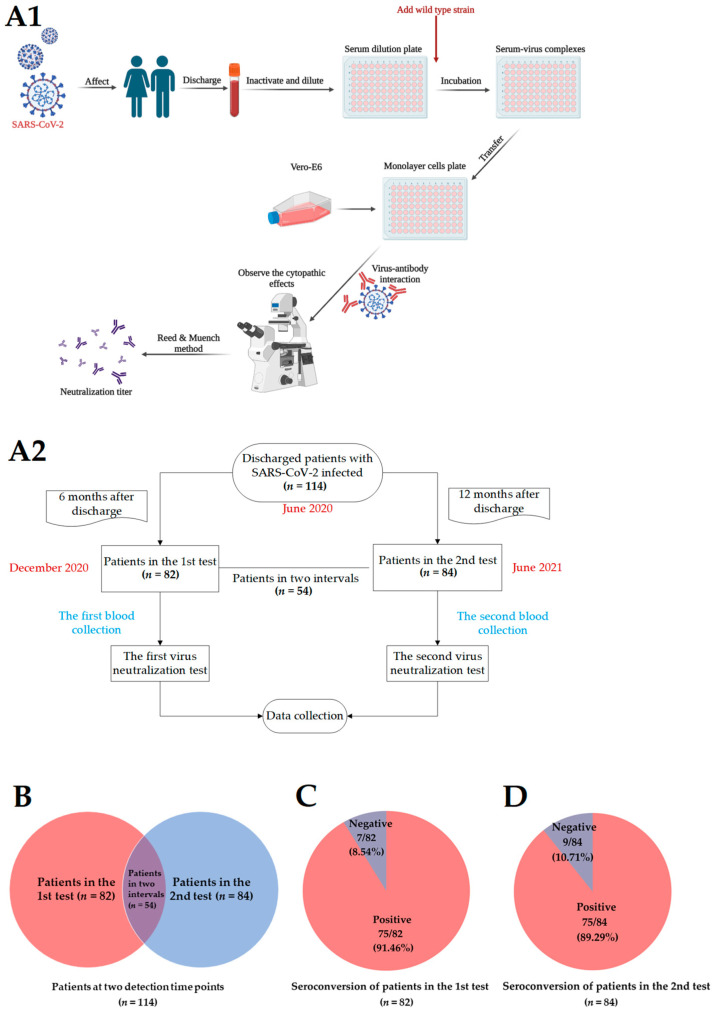
Experimental study scheme and anti-SARS-CoV-2 NAbs detection. (**A1**,**A2**) The experimental scheme of the entire study. (**B**) Patients at different time points. There were 114 patients, of whom 82 participated in the first test (6 months after discharge), 84 participated in the second test (12 months after discharge), and 54 participated in both tests. (**C**) NAbs seroconversion rates of the patients in the first test (*n* = 82). (**D**) NAbs seroconversion rates of the patients in the second test (*n* = 84). (**E**) The proportion of NAbs titer-positive at different time points after discharge by gender. Males (blue) and females (red). (**F**) The proportion of NAbs titer of patients at two detection time points after discharge (*n* = 54). SARS-CoV-2: Severe Acute Respiratory Syndrome Coronavirus 2; NAbs: neutralizing antibodies.

**Figure 3 pathogens-11-01531-f003:**
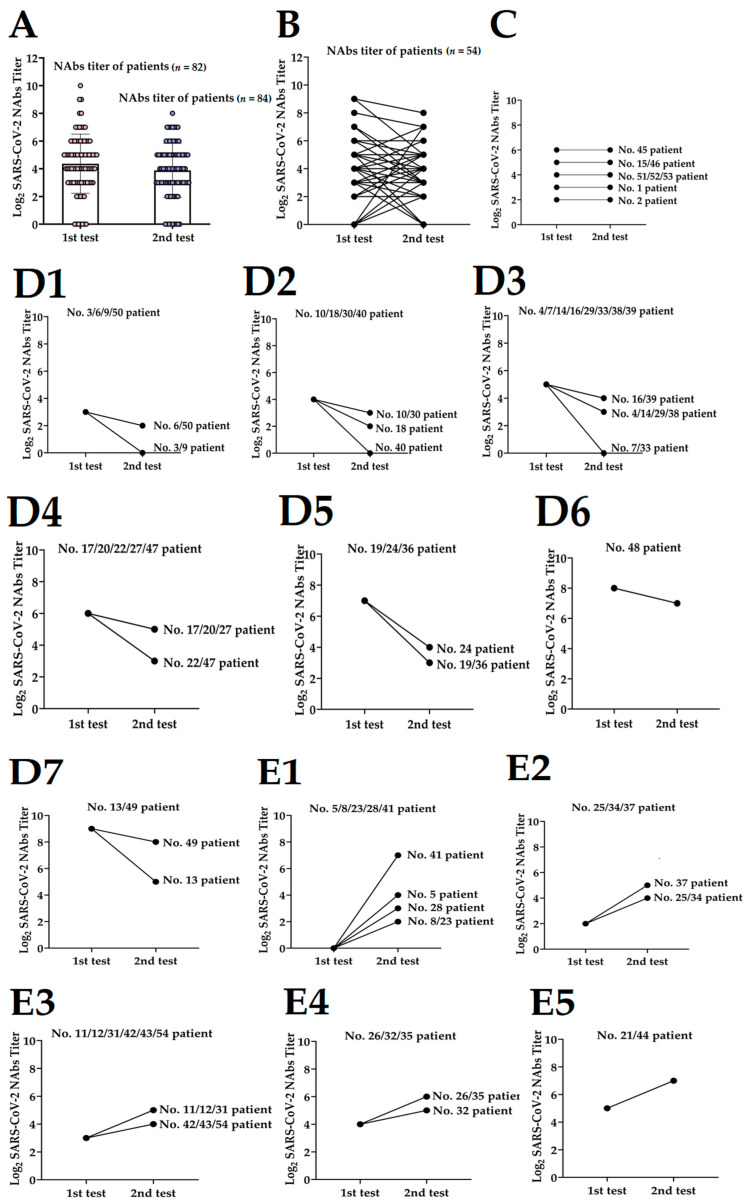
The overall distribution of NAbs titers of patients at different time points after discharge. NAbs titers were log_2_ processed. *p* values were determined with a two-tailed Student’s *t*-test. (**A**) The overall distribution of anti-SARS-CoV-2 NAbs titers of patients. Data were presented as median ± SEM. (**B**) The longitudinal dynamics of the anti-SARS-CoV-2 NAbs titers during follow-up (*n* = 54). (**C**) Dynamics changes in anti-SARS-CoV-2 NAbs titers of patients with an unconverted curve (*n* = 8). (**D1**–**D7**) Patients with a downtrend curve of NAbs titers levels (*n* = 27). (**E1**–**E5**) Partial patients with an uptrend curve of NAbs titers levels (*n* = 19). NAbs: neutralizing antibodies; SARS-CoV-2: Severe Acute Respiratory Syndrome Coronavirus 2; SEM: Standard Error of Mean.

**Figure 4 pathogens-11-01531-f004:**
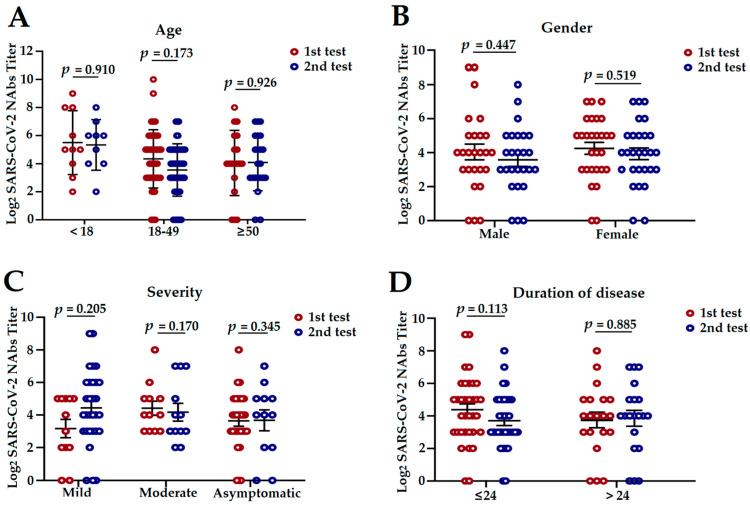
NAbs response on half a year and year after onset in 54 patients stratified according to age, gender, disease severity, and duration of disease; first test (red) and second test (blue). NAbs titers were log_2_ processed. *p* values were determined using a two-tailed one-way *ANOVA* test. (**A**) Comparison of NAbs titers of 54 patients stratified according to age; <18 years (*n* = 7), 18–50 years (*n* = 36), and ≥50 years (*n* = 11). (**B**) Comparison of NAbs titers of 54 patients stratified according to gender; male (*n* = 26) and female (*n* = 28). (**C**) Comparison of NAbs titers of 54 patients stratified according to disease severity; mild (*n* = 12), moderate (*n* = 30), and asymptomatic (*n* = 12). (**D**) Comparison of NAbs titers of 54 patients stratified according to the duration of disease; ≤24 days (*n* = 34) and >24 days (*n* = 20). Data were presented as median ± SEM. *p* values were determined by applying an *F* test. NAbs: neutralizing antibodies; *ANOVA*: analysis of variance; SEM: Standard Error of Mean.

**Figure 5 pathogens-11-01531-f005:**
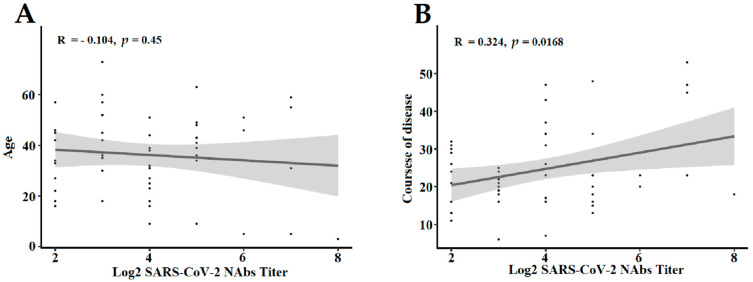
Correlation analysis of the Nabs titers and age and duration of the disease. NAbs titers were log_2_ processed. (**A**) Correlation between the Nabs titer and age. (**B**) Correlation between the Nabs titer and duration of the disease. Correlations were calculated by Pearson correlation coefficients. NAbs: neutralizing antibodies.

**Table 1 pathogens-11-01531-t001:** Demographic and clinical characteristics of 114 patients with COVID-19. COVID-19: Coronavirus Disease 2019.

	Male (*n* = 56)	Female (*n* = 58)	Total (*n* = 114)	*p*-Value
Age, years				0.192
<18	10	4	14	
18–50	33	37	70	
≥50	13	17	30	
Median age (IQR)	39.00 (27.00–48.75)	39.00 (29.75–52.25)	39.00 (27.75–50.25)	
Clinical classification				1.000
Asymptomatic case	9	9	18	
Mild case	7	9	16	
Moderate case	40	40	80	
NAbs titer				
1st test (*n* = 82)	41	41	82	1.000
Seronegative (<1:4)	4	3	7	
Seropositive (≥1:4)	37	38	75	
GMT_1st test_			1:23.21	
2nd test (*n* = 84)	39	45	84	0.292
Seronegative (<1:4)	6	3	9	
Seropositive (≥1:4)	33	42	75	
GMT_2nd test_			1:17.09	
Average duration of disease (days)	25.06	22.87	23.90	0.258

Note: COVID-19, Coronavirus Disease 2019; GMT, geometric mean titer.

**Table 2 pathogens-11-01531-t002:** Comparison of different NAbs-titer groups in gender, age, clinical classification, and duration of the disease. NAbs: neutralizing antibodies.

		Downtrend Curve ^1^	Uptrend Curve ^2^	Unconverted Curve ^3^	*p*-Value
Total		10	27	17	0.302
Male		7	12	7	
Female		3	15	10	
Age (years)		39.00 (30.00–49.00)	43.00 (31.50–51.00)	22.50 (16.00–36.25)	0.026 ^a*^
Clinical classification	Asymptomatic case	6	5	1	0.003 ^b*^
	Mild case	2	3	5	
	Moderate case	1	18	11	
Duration of disease		20.69	27.88	29.89	0.015 ^c*^

Note: NAbs, neutralizing antibodies. ^1^: represented the NAbs titers after one year were lower than those after half a year. ^2^: remarked the NAbs titers after one year were higher than those after half a year. ^3^: indicated the NAbs titers after one year were consistent with those after half a year. *: meant a statistically significant result. ^a^: For the age, the statistical differences in the NAbs titer groups were tested by One-way *ANOVA* test, and Bonferroni correction showed that there was a significant difference between the “Downtrend curve” and “Uptrend curve” (*p* = 0.010). ^b^: for clinical classification, the statistical differences in the NAbs titer groups were tested by Fisher‘s exact test, and through Bonferroni correction showed that there was a significant difference between the “Uptrend curve” and “Unconverted curve” (*p* = 0.002) and “Downtrend curve” (*p* = 0.001). ^c^: for the duration of the disease, the statistical differences in the NAbs titer groups were tested by One-way *ANOVA* test, and through Bonferroni correction showed that there was a significant difference “Downtrend curve” and “Uptrend curve” (*p* = 0.015).

**Table 3 pathogens-11-01531-t003:** Demographic characteristics, vaccination histories, and GMTs of 62 patients with COVID-19.

	S_Alpha_ (*n* = 3)	S_Beta_ (*n* = 3)	S_Delta_ (*n* = 31)	S_Omicron_ (*n* = 25)
Median Age, years	39	47	44	35
Gender				
Male	2	3	12	4
Female	1	0	19	21
Vaccination history				
Non-vaccinees	3	3	19	1
1st dose	0	0	7	1
2nd dose	0	0	5	9
3rd dose	0	0	0	14
GMTs				
V_Wild_	50.4	5.85	25.60	263.55
V_Alpha_	25.20	1.00	16.62	256.34
V_Beta_	31.75	7.94	12.06	164.50
V_Gamma_	80.00	40.00	87.48	572.82
V_Delta_	31.75	3.42	101.57	223.16
V_Omicron_	2.15	1.71	3.19	102.67

Note: GMT, geometric mean titer; COVID-19, Coronavirus Disease 2019.

**Table 4 pathogens-11-01531-t004:** The relationship between the GMTs and vaccination histories of S_Omicron_ and S_Delta_ from discharged patients.

Vaccination History	GMTs of S_Delta_ (*n* = 31)						Vaccination History	GMTs of S_Omicron_ (*n* = 25)					
	V_Wild_	V_Alpha_	V_Beta_	V_Gamma_	V_Delta_	V_Omicron_		V_Wild_	V_Alpha_	V_Beta_	V_Gamma_	V_Delta_	V_Omicron_
Non-vaccinees (*n* = 19)	18.84	24.32	19.54	115.22	160	4.78	Non-vaccinees (*n* = 1)	160	160	20	160	160	40
1st dose (*n* = 7)	63.57	5.72	3.38	24.38	23.61	1.26	1st dose (*n* = 1)	10	160	160	320	80	40
2nd dose (*n* = 5)	22.97	17.41	11.49	183.79	139.29	2.51	2nd dose (*n* = 9)	296.28	320	217.73	507.97	217.73	93.32
3rd dose (*n* = 0)	0	0	0	0	0	0	3rd dose (*n* = 14)	320	237.76	160	706.62	249.83	124.91

Note: GMT, geometric mean titer.

## Data Availability

Not applicable.

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
