# Peer review of "Progress of the COVID-19: Persistence, Effectiveness, and Immune Escape of the Neutralizing Antibody in Convalescent Serum"

_pathogens, 2022, doi:10.3390/pathogens11121531_

Round 1

Reviewer 1 Report

The study investigated levels of neutralizing antibodies (Nabs) against a SARS-CoV-2 strain (wild type: No.20SF014) in COVID-19 patients at two time points (6 and 12 months). The following are suggested to improve this paper:

1. Methods:

 - Please define how cytopathogenic effect is detected in the study, as well as the number of replicates. Why is a luciferase assay not performed? 

- Indicate the cutoff titer used for seropositivity (1:4?).

- Indicate the different influencing variables/factors and how they are determined (age, sex, course of the disease).

- For the course of disease, please state the reference for using 24 days as cutoff. Also, define this parameter better (is it duration of symptoms or positivity for the antigen/virus?)

2. Results

- In lines 157-158, please indicate the P value of the difference between the 1st and 2nd tests (as in line 172). Is this sentence redundant? If so, please state this in one section only.

- In line 234, what does moderate correlation and negative correlation mean? Please elaborate and use P values to determine significance.

- In Table 2, Total (male/female) can be presented as male and female in different rows, and total can be displayed as part of column headings.

- In Figure 2, legends are too small to read. Please separate figures A and B. Figure C may not be needed. For figures D and E, the authors can present the average results of these patients rather than present 5 individual patient graphs.

- In line 201, please define what the authors mean by "decrease negative"

3. Conclusion - This section should also enumerate the different influencing factors that affect NAbs titer.

Minor Comments:

1. Consider replacing "1st" and "2nd" to 6-month and 12-month test, respectively, since not all patients had both testing done.

2. Consider changing "course of disease" to "duration of disease/symptoms." Please clarify and define this better.

3. Please review and correct the multiple grammatical and spelling errors.

Overall, this paper has potential but should consider applying major changes as above. 

Author Response

Dear Editors and Reviewers,

 We are very grateful to all Reviewers for reviewing our manuscript entitled “The dynamics of neutralizing antibody level in one year after SARS-CoV-2 infection” (ID: pathogens-2032200). Those comments are all valuable and very helpful for revising and improving our paper, as well as the important guiding significance to our researchers. We have studied the comments carefully and have made corrections which we hope meet with approval as follows. Revised portions are tracked in the paper and the point-to-point responses to the reviewers’ comments are marked in red. Please see the attachment.

Best, 

Liang Dan

Reviewer 2 Report

The authors characterized the kinetics of neutralizing antibody (NAbs) titers up to one year after SARS-CoV-2 infection by a micro-neutralization assay for 114 patients. Methodology looks correct and the manuscript is well written, however the authors only measured NAbs antibody against Wuhan original strain but not other strains. I think the authors need to test against current strain strains to attract readers. Comments for the authors:

Major points:

1.    Please describe which strain was epidemic between June 2020 and June 2021.

2.    Please indicate vaccination history.

3.    Please include Nabs titers against Omicron.

Minor points:

1.    Please increase the font size in the Figures. Currently very difficult to read.

Author Response

(The authors gave the same response as above.)

Round 2

Reviewer 1 Report

Thank you for making the changes and adding additional experiments/clinical information. There are still numerous grammatical errors in the document. Other than these, the paper has significantly improved.

Author Response

Dear Reviewer,

We have made detailed revisions to the manuscript and hope you will reconsider our response.

All in all, special thanks to you for your good comments.

Best,

Liang Dan

Reviewer 2 Report

Resubmitted manuscript by L. Dan et al., explores the fate of Nab in convalescent patients of COVID-19. The authors have responded extensively and in full to previously raised concerns and suggestions. A few issues remain:

1.    P.3, Line5: Please delete “Additionally”.

2.    P.5, Last Line: “Figure 2A1-1A2” should be “Figure 2A1-2A2”

3.    P.7, Figure 2F: Positive should be 49/54.

4.    P.14, Table 3: I only see 62 patients but not 75.

5.    P.14: Figure S2 is missing.

Author Response

(The authors gave the same response as above.)
